# Dietary Polyphenols Targeting Arterial Stiffness: Interplay of Contributing Mechanisms and Gut Microbiome-Related Metabolism

**DOI:** 10.3390/nu11030578

**Published:** 2019-03-08

**Authors:** Tess De Bruyne, Bieke Steenput, Lynn Roth, Guido R. Y. De Meyer, Claudia Nunes dos Santos, Kateřina Valentová, Maija Dambrova, Nina Hermans

**Affiliations:** 1Laboratory of Natural Products and Food-Research and Analysis (NatuRA), University of Antwerp, 2610 Antwerpen, Belgium; tess.debruyne@uantwerpen.be (T.D.B.); bieke.steenput@uantwerpen.be (B.S.); 2Laboratory of Physiopharmacology, University of Antwerp, 2610 Antwerpen, Belgium; lynn.roth@uantwerpen.be (L.R.); guido.demeyer@uantwerpen.be (G.R.Y.D.M.); 3Instituto de Biologia Experimental e Tecnológica, Apartado 12, 2780-901 Oeiras, Portugal; claudia.nunes.santos@nms.unl.pt; 4Instituto de Tecnologia Química e Biológica, Universidade Nova de Lisboa, Av. da República, 2780-157 Oeiras, Portugal; 5CEDOC, NOVA Medical School, Faculdade de Ciências Médicas, Universidade Nova de Lisboa, Campo Mártires da Pátria, 130, 1169-056 Lisboa, Portugal; 6Institute of Microbiology of the Czech Academy of Sciences, Vídeňská 1083, 142 20 Prague, Czech Republic; kata.valentova@email.cz; 7Laboratory of Pharmaceutical Pharmacology, Latvian Institute of Organic Synthesis, LV-1006 Riga, Latvia; maija.dambrova@farm.osi.lv

**Keywords:** arterial stiffness, aging, polyphenols, polyphenol biotransformation, gut microbiome, antioxidant, anti-inflammatory, autophagy

## Abstract

Increased arterial stiffness is a degenerative vascular process, progressing with age that leads to a reduced capability of arteries to expand and contract in response to pressure changes. This progressive degeneration mainly affects the extracellular matrix of elastic arteries and causes loss of vascular elasticity. Recent studies point to significant interference of dietary polyphenols with mechanisms involved in the pathophysiology and progression of arterial stiffness. This review summarizes data from epidemiological and interventional studies on the effect of polyphenols on vascular stiffness as an illustration of current research and addresses possible etiological factors targeted by polyphenols, including pathways of vascular functionality, oxidative status, inflammation, glycation, and autophagy. Effects can either be inflicted directly by the dietary polyphenols or indirectly by metabolites originated from the host or microbial metabolic processes. The composition of the gut microbiome, therefore, determines the resulting metabolome and, as a consequence, the observed activity. On the other hand, polyphenols also influence the intestinal microbial composition, and therefore the metabolites available for interaction with relevant targets. As such, targeting the gut microbiome is another potential treatment option for arterial stiffness.

## 1. Introduction

Increased arterial stiffness is a degenerative vascular aging process, which mainly affects the extracellular matrix of elastic arteries and results in a reduced capability of arteries to expand and contract in response to pressure changes [1,2,3,4]. Stiffening of the arterial wall leads to fundamental changes in central hemodynamics with the increased pulsatile strain on the microcirculation, leading to detrimental consequences for end organ function [5,6]. Arterial stiffness is an independent risk factor for cardiovascular (CV) morbidity and mortality [5,7,8].

Etiological factors for the development of arterial stiffness include pathways of vascular functionality, oxidative status, inflammation, glycation, and autophagy. The most relevant activities of dietary polyphenols on mechanisms involved in the pathophysiology and progression of arterial stiffness (AS) and its consequences are identified and discussed.

Polyphenol activities are not only due to direct effects of the parent compounds but are also largely dependent on the bioactivity of their metabolites, originating from the host or microbial metabolic processes. The composition of the gut microbiome, therefore, determines the resulting metabolome and the observed effect. Moreover, polyphenols also modulate the intestinal microbial composition and, therefore, play a role in determining the metabolites available for interaction with relevant targets. Thus, targeting the gut microbiome is another potential treatment option for arterial stiffness.

## 2. Arterial Stiffness

Normal elastic arteries have a smoothing function, thus assuring a steady blood flow in peripheral tissues [9]. The collagen/elastin ratio determines the stiffness of the vascular wall. However, aging and other risk factors cause the arterial tissue to gradually lose its elasticity, primarily due to progressive degeneration of the extracellular matrix in the media layer, which leads to arterial stiffness [2,4,8].

The decrease in the elastin/collagen ratio in the media layer progresses under the influence of altered lysyl oxidase (LOX) and matrix metalloproteinase (MMP) activity [5]. Elastin is progressively fragmented and degraded, while the amount of collagen increases, and also collagen-elastin cross-links multiply under the influence of *S*-nitrosylation and advanced glycation end-products (AGEs). Angiotensin II (Ang II) signaling further contributes to collagen and advanced glycation endpoints accumulation and elastin degradation.

Additionally, upon activation of the receptor of AGE (RAGE), intracellular reactive oxygen species (ROS) levels are increased through upregulation of reduced nicotinamide adenine dinucleotide (phosphate) (NAD(P)H) oxidase expression, thus contributing to the oxidative stress associated with arterial stiffness. Moreover, calcium microdisposition in fragmented elastin leads to vascular calcification, which further increases the stiffening process [5,8,9,10,11,12,13].

Stiffened arteries contribute to increased systolic blood pressure (SBP), amplified by superposition of prematurely reflected pulse waves [12]. On the other hand, increased blood pressure (BP) is also a cause for the reduction of arterial elasticity. Therefore, the unraveling of contributing mechanisms is complicated considerably [14]. High SBP primarily affects microvasculature in vulnerable end-organs like the brain and kidney [12,15]. Raised SBP also requires increased left ventricular (LV) workload and, as a consequence, there is a need for increased coronary perfusion and oxygen. However, the diastolic pressure determining coronary flow is insufficient, causing LV dysfunction, hypertrophy, and coronary ischemia, which are known risk factors for CV events [1,2,5,12].

## 3. Factors Contributing to Arterial Stiffness

**Aging** and related processes are the main determinants of stiffness in elastic arteries, which are correlated with nutritional and lifestyle factors and subsequent age-associated disorders, such as metabolic syndrome, type 2 diabetes, hypertension, atherosclerosis, and renal disease, thus implying metabolic factors in its pathogenesis [10,11,16,17]. Degeneration and remodeling of elastic components of the arterial wall usually become important after the age of 55, concomitant with a decrease in intracellular magnesium [17]. Furthermore, calcification, apoptosis, inflammation and oxidative and nitrosative stress, genetic influences, as well as reduced autophagy, play a role in age-related stiffening [5,11,16,17,18].

**Nutrition and lifestyle in general** influences are important for protection against the development of arterial aging. Generally, diets and nutrients that reduce oxidative stress and inflammation, such as diets high in fruits and vegetables, grains, nuts, seeds, legumes, low-fat meat, and fish, with limited amounts of refined foods are associated with reduced arterial stiffness [19,20]. Specific nutrient measures, such as restricted dietary salt, or specific foods, such as dairy products, fermented dairy, dark chocolate, tea, soy, olive oil, grains, and nuts, have been shown to have some beneficial effects. Energy intake restriction and aerobic exercise also protect against arterial aging [9,17,19]. Smoking, on the other hand, has an adverse effect on arterial stiffness [17]. Extreme (both long and short) sleep duration and poor sleep quality are associated with enhanced arterial stiffness and are possibly linked to increased MMP expression [5]. Mental stress also contributes to vascular dysfunction involving oxidative stress and inflammation [21].

**Metabolic syndrome**-related medical conditions induce vascular stiffening. **Hypertension** can lead to arterial stiffening through the upregulation of pathways involved in inflammation, fibrosis, and wall hypertrophy. **Diabetes**-accelerated arterial stiffening includes elevated levels of oxidative stress, similar to age-induced stiffness, MMP-mediated elastin fragmentation, and calcification, and **obesity** results in aortic stiffening, at least in part mediated through LOX-downregulation, leading to elastin fragmentation. Modest weight loss results in the improvement of arterial stiffness [5,22].

**Oxidative stress** is an important factor in the development of arterial stiffness. During vascular oxidative stress, enzymatic systems in the vascular wall, including NADPH oxidase, mitochondrial enzymes, dysfunctional endothelial nitric oxide synthase (eNOS), and xanthine oxidase (XO), produce ROS. Antioxidant systems that counteract ROS generation include superoxide dismutase (SOD), catalase (CAT), glutathione peroxidase (GPx), paraoxonase (PONs), thioredoxin (TRX) peroxidase, and heme oxygenase (Hmox) [23]. Elevated levels of superoxide radical anion react with nitric oxide (NO) to produce peroxynitrite. Oxidative and nitrosative stress contribute to arterial stiffness etiology due to oxidative damage to lipids, proteins, and DNA in endothelial cells and uncoupling of NO synthase, leading to endothelial dysfunction. Moreover, altered blood flow also increases ROS production, and mitochondrial oxidative stress and SOD 2 deficiency induce aortic stiffening [5]. Stiff arteries potentially induce a positive feedback mechanism that downregulates eNOS and upregulates endothelin-1 (ET-1), thus further increasing wall stiffness [10]. Several markers of oxidative stress have been associated with increased arterial stiffness, although a causative link has sometimes been questioned due to the experimental complexity of antioxidant clinical trials. Nevertheless, changes in malondialdehyde (as thiobarbituric acid reactive substances, TBARS), SOD, vascular adhesion protein-1, oxidized low-density lipoprotein (oxLDL), and isoprostanes have been reported in vascular stiffness [24].

**Inflammation** is involved in arterial stiffness development by impairment of endothelial function by reducing NO bioavailability and increasing ET-1 [10]. The contribution of an inflammatory status is furthermore reflected in the role of MMPs in elastin degradation, the overexpression of lectin-like oxidized low-density lipoprotein receptor 1 (LOX-1, by a nuclear factor kappa B (NF-κB) dependent mechanism), increasing uptake of oxLDL, the transdifferentiation of vascular smooth muscle cells (VSMCs) into an osteoblastic phenotype under inflammatory conditions, influence of cytokines, increased AGE’s synthesis, C-reactive protein (CRP: inhibits endothelial NO synthase, increases cytokine expression and generation of ROS, affects adhesion molecules and microRNAs) [25]. Inflammation can, therefore, induce functional and structural arterial stiffening [26]. Chronic low-grade inflammation interacts synergistically with oxidative stress, but the order and relationship between these events are unclear [19].

Currently, limited evidence is available about the possible role of age-related impaired **autophagy** in arterial stiffness and endothelial dysfunction. This process is associated with increased levels of oxidative stress and inflammation [18,27]. ROS can induce autophagy as a defense mechanism against cell death [28], but recent evidence also indicates that autophagy may be involved in antioxidant defense mechanisms by taking part in the reduction and repair of oxidative damage [29], which could have a strong impact on cardiovascular health.

Heritability of arterial stiffness is about 40%. Associations of **gene** expression levels with arterial stiffening have been found for genes involved in extracellular matrix and calcification and for genes involved in blood pressure regulation. However, still very little is known about the molecular mechanisms underlying phenotypic variability [8]. Candidate genes (e.g., insulin-like growth factor 1 (IGF-1) receptor, interleukin 6 (IL-6), proprotein convertase PACE4, etc.) potentially involved in arterial elasticity have been found on chromosomes 2,7,13, and 15 [17].

Factors contributing to the pathophysiology of arterial stiffness have been summarized in Figure 1.

## 4. Dietary Polyphenols

Polyphenols are a widespread class of plant secondary metabolites with a diverse range of biological activities. About 8000 polyphenolic structures have been identified, which can be subdivided according to their chemical structure into the following structural classes: phenolic acids, lignans, stilbenes, flavonoids (including isoflavonoids and anthocyanins), condensed and hydrolyzable tannins. They are found in several foods: the dietary polyphenols most investigated for their vascular properties include flavonoids (flavanols) and procyanidins in chocolate (*Theobroma cacao*), catechins, such as epigallocatechin gallate (EGCG) in green tea (*Camellia sinensis*), isoflavones in soy (*Glycine max*), curcumin from turmeric (*Curcuma longa*), oleuropein and hydroxytyrosol (HT) in olives (*Olea europaea*), anthocyanins in berries, resveratrol and other stilbenes in grapes and wine (*Vitis vinifera*) [30]. Main polyphenol classes, major compounds in each class, and important dietary sources are summarized in Table 1 [31].

In general, epidemiological studies and interventional trials suggest an inverse association between dietary polyphenol intake and cardiovascular events both in the general population and in patients with cardiovascular risk factors [32,33,34].

In view of the etiology of arterial stiffness, and the important contribution hereto of oxidative and nitrosative stress and inflammation, among others, plant polyphenols may potentially be effective biological agents for reducing the occurrence and progression of stiffening arteries. Evidence for the effects of food polyphenols on arterial stiffness is, however, rather limited due to the scarcity and heterogeneity of study designs in interventional trials, the complexity of observational trials, and the problems related to the translation of observations from animal models to human subjects. Moreover, there are important difficulties in assessing polyphenol intake, and there is a lack of uniformity in biomarkers and endpoints [35]. Nevertheless, limited relevant data exist, and known interference of polyphenols with mechanisms involved in arterial stiffness allows the identification of promising interactions.

## 5. Bioavailability of Polyphenols

The biological activity of dietary polyphenols depends on their bioavailability, intestinal absorption, and metabolism in the gastrointestinal tract, which itself depend on their chemical structure. Polyphenols can be absorbed from the small intestine, but, more often, as they are frequently present as esters, glycosides or polymers in their food matrix, they cannot be absorbed as such. To be absorbed, these molecules must first be hydrolyzed by intestinal enzymes or by the colonic microbiota. Phase II metabolism then converts them to methylated, sulfated and/or glucuronidated metabolites [36] (Figure 2). Polyphenols are thus rapidly degraded and metabolized and are often poorly absorbed resulting in limited bioavailability. Most native polyphenols are only found in nM to low µM ranges, and in plasma, glucuronidated, sulfated, and methylated derivatives are found in addition to the free phenolic form [37,38].

On the other hand, large differences in bioavailability exist; for example, some flavonoid classes could be sufficiently absorbed to exert cardioprotective effects in vivo [15,39,40,41]. Furthermore, the intracellular deconjugation metabolism of phase II metabolites has to be taken into account, releasing parent polyphenols in cells and tissues and provoking local activity.

## 6. Polyphenols and the Intestinal Microbiome

### 6.1. Polyphenol Metabolism by the Gut Microbiota

A significant amount of polyphenols, including conjugated metabolites from the bile (90–95% of the total polyphenols intake), reaches the colon [43]. Subsequently, they are transformed by gut microbiota enzymes into a wide range of low-molecular phenolic compounds, which may act as the true bioactive agents [38]. The crucial role of gut microbiota for polyphenol metabolism has been highlighted initially by studies showing that germ-free animals did not form phenolic acid metabolites from flavonoids [44].

The microbial composition of the intestine, therefore, has great relevance for the individual response to these compounds. Core intestinal microbiota are relatively stable in adulthood and are dominated by the *Firmicutes* and *Bacteroidetes*, and to a lesser extent *Actinobacteria*, *Proteobacteria,* and *Verrucomicrobia*. Classification of human-associated bacteria into three enterotypes, depending on the dominating bacterial genus, has been proposed [45]. However, lifestyle and diet can induce major changes: an acute change in diet alters microbial composition within 24 h [46,47]. Moreover, aging, a determining factor in AS, induces important microbial modifications, such as lower diversity, a shift in dominant species, a decrease of beneficial species, lower bioavailability of short-chain fatty acids (SCFA), and a greater inter-individual variation [45]. Polyphenol metabolism requires a consortium of microbes [48]. For some common dietary polyphenols, the prevalent gut microbial metabolites have been identified [49,50,51]; nevertheless, large inter-individual variations occur, which is attributed to differences in gut microbiota [48]. Differing metabolomes result in varying bioactivity and polyphenol health benefits [39,48]. Heterogeneity in responsiveness obscures associations between intake and potential health benefits [39].

Interindividual heterogeneity is, for instance, illustrated in the biotransformation of some isoflavones into *S*-equol (Figure 3A), which is thought to have higher efficacy than the parent compound. Only about 30% of the Western population and 60% of Asian subjects can produce equol and, therefore, have more beneficial health effects from soy consumption, due to the presence of specific bacteria in the gut [30,52]. Equol producers were shown to have significantly lower triglyceride (TG) and carotid intima thickness levels compared to non-equol excretors in a study involving 572 Chinese participants [53]. On the other hand, while soy intake improved carotid-femoral pulse wave velocity (PWV) in equol producers, a single dose of *S*-equol displayed no cardiovascular benefits in non–equol excretors suggesting a more complex role of the equol producer phenotype [39,54]. Equol-producing bacteria include species of the family Coriobacteriaceae, which are associated with beneficial properties in obesity and diabetes [55]. The occurrence of these bacterial species of the genera *Adlercreutzia*, *Eggerthella*, *Paraeggerthella*, and *Slackia* could, therefore, be a potential biomarker for a beneficial response to the consumption of flavonoids in cardiometabolic diseases [39].

Bacterial conversion of flavan-3-ol monomers, such as catechin, epicatechin, gallocatechin, epigallocatechin, and their corresponding gallate esters in the human intestine, includes the hydrolysis of ester bonds, the reductive cleavage of the C-ring, and further conversion of the resulting 1,3-diphenylpropan-2-ols to the corresponding γ-valerolactone and valeric acid (Figure 3B) [56,57]. Urinary excretion of γ-valerolactones was found to be lower in elderly (70 ± 4 y) compared to young (26 ± 6 y) subjects, which may influence the impact of, for example, cocoa flavan-3-ol consumption on arterial stiffness and related cardiovascular conditions [58].

Ellagitannins are biotransformed by gut microbiota into ellagic acid, which is then subject to the lactone ring opening and decarboxylation by *Gordonibacter* strains from Coriobacteraceae resulting in the formation of urolithin M5. Urolithin M5 is further transformed by dehydroxylation through various intermediates to urolithin A and urolithin B (Figure 3C), depending on the composition of the gut microbiome [59]. Three metabotypes (A, B, and 0) have been described. The B metabotype, which produces urolithin-B as the main metabolite, is more prevalent in overweight individuals, patients with metabolic syndrome or patients with colorectal cancer than in healthy individuals. It was also suggested that metabotype B individuals were at higher cardiovascular disease risk than metabotype A subjects (urolithin A producers) [60]. Aging was recently found to be the main factor determining the urolithin metabotypes in a Caucasian cohort of 839 subjects [61].

Enterodiol, a metabolite of lignans, may undergo dehydrogenation (cyclization) by *Lactonifactor longoviformis* leading to the formation of enterolactone (Figure 3D), which is known for its beneficial biological activities. High-producers of enterolactone from lignans have a lower risk of type 2 diabetes, and high serum enterolactone level was found to be associated with reduced coronary heart disease and CV disease-related mortality in middle-aged Finnish men [62,63]. High frequent dietary intake of lignans was found to be associated with decreased aortic PWV in postmenopausal and especially older women [64].

Another example is rutin, quercetin-3-*O*-rutinoside, that is, quercetin-3-*O*-(6-*O*-α-l-rhamnopyranosyl)-β-d-glucopyranoside, a component of drugs and food supplements intended to strengthen and increase the flexibility of blood vessels in bruises, spider veins, varicose veins, and hemorrhoids, but also to reduce post-thrombotic syndrome, venous insufficiency or endothelial dysfunction. Similarly to other polyphenols, there is a lack of high-quality scientific evidence from clinical trials for rutin efficacy, possibly partially due to its low bioavailability [65,66]. As no human-l-rhamnosidase or rutinosidase exists, rutin, like other flavonoids conjugated with rhamnose moiety (e.g., hesperidin, naringin), cannot be hydrolyzed in the small intestine and the aglycone can only be released from rhamnose moiety (Figure 3E) by microbial enzymes in the gut [67]. Moreover, not all intestinal strains possess rhamnose cleaving ability, and in some people, such enzymes seem to be lacking [68,69]. The aglycone quercetin is further extensively metabolized by microbial enzymes to an array of smaller phenolics. Some of them, such as 3-(3-hydroxyphenyl) propionic and 3,4-dihydroxyphenylacetic acid and 4-methylcatechol, displayed vasodilatory activity both in vitro and in vivo [70,71]. Microbial biotransformation of isoflavonoids, flavan-3-ols, ellagitannins, lignans, and flavonoid rutinosides is summarized in Figure 3.

### 6.2. Polyphenols Modulate Intestinal Microbiota Composition

Regular consumption of polyphenol-rich foods could in turn influence the colonic bacterial populations and their metabolic activities, increasing inter-individual bioavailability variation. Consumption of polyphenols leads to a gut microbiome that is enriched in bacterial genera, such as *Lactobacillus* and *Bifidobacteria* [45,46,47]. Polyphenols can, therefore, exhibit a prebiotic-like effect and can potentially be used to selectively modulate the intestinal microbiome. The different microbial composition is translated into a significant difference in bacterial metabolite profiles, as illustrated in regular cocoa product consumers in comparison with non-consumers after dark chocolate intake [72]. Influence on the microbial composition has also been demonstrated in pigs and in rat studies [73,74]. Also, for grapes [75,76,77,78], apples [51], green tea and oolong tea polyphenols [79,80], blueberries [81], and extra virgin olive oil [82], modulation of the intestinal microbiome has been reported. Most studies have been carried out in rodent models, but also human trials are available.

There is thus a bidirectional phenolic—microbiota interaction. Stratification in clinical trials according to metabotypes is, therefore, necessary to fully assess the biological activity of polyphenols [60,83]. The complexity of the metabolic output of the gut microbiota, dependent to a large extent on the individual metabolic capacity, emphasizes the need for assessment of functional analyses using metabolomics in conjunction with the determination of gut microbiota composition [84].

### 6.3. Intestinal Microbial Metabolism with Impact on Cardiovascular Health

Besides the mutual interaction between polyphenols and the gut microbiome, additional microbial modulation of cardiovascular risk further complicates the interpretation of experimental and epidemiological data. Indeed, the gut microbiota derived metabolite trimethylamine-*N*-oxide (TMAO) has been implicated in the pathogenesis of cardiovascular diseases [85]. TMAO is formed from trimethylamine, which is synthesized by several intestinal bacteria from choline, betaine, and carnitine. Notably, *Firmicutes* and *Proteobacteria* are involved in its generation. TMAO induces vascular inflammation through mitogen-activated protein kinase (MAPK) and NF-κB signaling [86,87,88]. In a rat study, age-dependent dysbiosis was reflected in higher TMAO levels, resulting in vascular inflammation and oxidative stress, inhibition of eNOS and subsequent lower NO availability and endothelial dysfunction [87].

In a study in mice, seven months of a Western diet caused gut dysbiosis, increased arterial stiffness and endothelial dysfunction, and reduced *N*(ω)-nitro-l-arginine methyl ester (l-NAME)-inhibited dilation. Impairments in vascular function correlated with reductions in *Bifidobacterium spp*. Upon antibiotic treatment suppressing the gut microbiota, Western diet-induced arterial stiffness and endothelial dysfunction had been reversed [89].

The fact that the microbial community can be modulated by polyphenols has consequently an impact on arterial function and TMAO production. Resveratrol reduced the *Firmicutes/Bacteroidetes* ratio, and increased the genera *Lactobacillus* and *Bifidobacteria,* with subsequent lower TMAO levels in mice. Also, quercetin and apple procyanidins decreased the *Firmicutes/Bacteroidetes* ratio in rat and mouse models [88,90]. Recently, a possible role for some (poly)phenol-rich dietary products on the modulation of trimethylamine colonic production has been reported [91], suggesting colonic fermentation of carbohydrates as a mechanism to reduce trimethylamine and TMAO formation.

Moreover, a recent report describes the inverse association of AS (through PWV) with gut microbiome diversity in women, after adjustment for metabolic syndrome-related factors (lifestyle risk factors, cardiovascular risk factors, inflammatory markers, and metabolic factors). Microbial factors could explain 4–8% of the variance of PWV. Authors hypothesized that chronic endotoxemia and subsequent low-grade inflammation could be determining factors. Targeting the microbiome may be a valuable alternative for the treatment of arterial stiffness [92,93]. Indeed, soy consumption was shown to ameliorate inflammation and arterial stiffness, while decreasing the *Firmicutes*/*Bacteroidetes* ratio, in ovariectomized rats [94].

Due to extensive metabolism and complex relationship between polyphenols and the intestinal microbiome, many conflicting results have been reported between in vitro and in vivo studies. Although in vitro reports of observed activities are not always reflected in a clinical result, human in vivo activities do occur, often by direct interactions with receptors, enzymes, and signaling pathways, by modulation of gene expression through activation of various transcription factors or by the activity of degradation products and metabolites. For flavanols, a consensus has been obtained for their biological functions, which can occur at cellular and systemic levels by modulating cellular signaling and enzyme activities at intakes achievable with a normal diet. Randomized, controlled trials demonstrate an effect on blood pressure, LDL cholesterol, and flow-mediated dilation (FMD) [95].

It must be noted that the enormous diversity of chemical structures—parent and hydrolyzed polyphenols and phase II and microbial metabolites—hampers the identification of the active compound(s). Often, several metabolites or a whole array of related compounds could be responsible for the observed effects [96,97]. This review is focused on the effects of dietary polyphenols on AS, although mechanisms involved could also influence other cardiovascular pathologies. Some observational and interventional studies are cited as an illustration of current research, but care should be taken that great variability exists in compound/food studied, dosage, population, sample sizes, endpoints, and follow-up. Hereafter, rather than discussing effects of separate polyphenolic compounds or the foods that contain them, effects will be grouped according to the mechanism involved, keeping in mind that the same compound can display diverse activities and that several mechanisms, such as antioxidant and anti-inflammatory activities, are interrelated. Influencing many different targets with lower affinity, as polyphenols do, may result in a combined effect, which could be sufficient to provide an overall health benefit [98]. To our knowledge, this fills a gap of recent reviews on the effect of dietary polyphenols on AS. Besides a review on flavonoids and AS, a little overview of current evidence is available. Data discussed therein seem to support the improvement of AS with increased flavonoid intake [15]. Together with the mutual interaction of gut—polyphenol, potential beneficial effects of polyphenols in AS illustrate the complex management of vascular function in the elderly.

## 7. Epidemiological Studies with AS Assessment

Several epidemiological studies indicate a positive correlation between polyphenol-rich food intake and several cardiovascular endpoints, such as cardiovascular mortality, myocardial infarction, chronic heart disease, and heart failure. Studies specifically linking AS outcomes to polyphenols are heterogeneous and limited, but seem to indicate a beneficial effect. Besides the already mentioned limitations of dietary polyphenol trials on vascular stiffness, primarily a correct estimation of polyphenol intake (usually using food frequency questionnaires linked to databases with dietary polyphenol contents) seems difficult, especially that of non-extractable polyphenols, which is a very important fraction contributing to total polyphenol intake [35]. Higher polymers with larger molecular weights will not be bioavailable as such, but their metabolites could contribute to the observed effects. Assessing polyphenol intake should be addressed by the development and measurement of adequate robust and validated biomarkers, either individual compounds or a panel of compounds, in plasma or urine; for example, 5-(3′,4′-dihydroxylphenyl)-γ-valerolactone is a suitable nutritional biomarker for the estimation of flavan-3-ol intake [35,99].

Higher anthocyanin and flavone, as well as a higher cocoa intake, have been linked to lower PWV [15,100,101]. Also, phytoestrogens (isoflavones and lignans) reduced PWV [64]. Epidemiological studies on soy isoflavones, in general, demonstrated improved arterial compliance, induced nitrite/nitrate levels, and decreased ET-1 levels in men and postmenopausal women [102].

## 8. Interventional Human Studies with AS Assessment

Several, usually small scale, interventional studies on dietary polyphenols and AS have been published. These studies are heterogeneous in population, dose, markers, and follow-up. The relevance of generally small AS effects observed for clinical outcomes remains to be investigated. Often, evaluation of AS markers is combined with the registration of effects on BP and endothelial function. All those parameters largely influence each other or are subject to modulation by inflammation and oxidative stress. The most evidence exists for the beneficial effect of cocoa and its derived products.

### 8.1. Cocoa, Coffee, Tea, and Their Isolated Polyphenols

Cocoa and chocolate are rich in flavonoids and proanthocyanidins. Several publications exist on the reduction of blood pressure upon cocoa consumption, with generally a more pronounced effect on SBP than in diastolic blood pressure (DBP). Cocoa intake has also been associated with decreased cardiovascular risk. Numerous studies also report improvement in vascular function, measured by brachial FMD, PWV or aortic augmentation index (Aix). Effects were best correlated with flavanol intake and plasma concentrations. The positive effect of cocoa flavonoids has been observed in healthy individuals, as well as in hypertensive, diabetic, obese, cardiovascular or renal disease patients. Potential mechanisms include activation of NO synthase, increased bioavailability of NO, antioxidant and anti-inflammatory properties [33,103]. For the isolated flavonoids, conflicting results have been reported. As an illustration, a few recent reports on reduction of arterial stiffness with cocoa or its derived products are listed in Table 2.

Although some deviating results have been reported, the bulk of evidence observed in both epidemiological and interventional studies led to the approval of a health claim about the effect of cocoa polyphenols on maintaining blood vessel elasticity, by the European Food Safety Authority (EFSA). To achieve this, 200 mg of cocoa flavanols, consumed as 2.5 g high-flavanol cocoa powder or 10 g high flavanol dark chocolate, should be ingested daily [104]. Simultaneous administration of methylxanthines (theobromine, caffeine) with cocoa (polyphenols) resulted in a more pronounced effect on brachial PWV and on FMD and circulating angiogenic cells. This was associated with increased plasma concentrations of (−)-epicatechin metabolites, suggesting an increased absorption [105]. Apparently, caffeine did not show comparable effects on coffee polyphenols [103].

Coffee polyphenol extract, with chlorogenic acids as main polyphenols, as well as isolated chlorogenic acids, improved FMD in two small trials (Table 2). Black tea (flavonoids, theaflavins, thearubigins) intake has been shown to decrease dose-dependently AS and BP in healthy volunteers, while green tea (mainly flavanols, like epigallocatechin-3-gallate) failed to show any alterations in PWV or in inflammatory markers in type 2 diabetes (T2D) patients. Genotype differences regarding catechol-*O*-methyltransferase (COMT) have been reported in a study with green tea (Table 2). In contrast to cocoa, the counteracting effect of caffeine in tea reduced the potential beneficial effect of the tea polyphenols [103].

### 8.2. Fruit, Wine, and Their Isolated Polyphenols

In general, men with increased cardiovascular disease (CVD) risk that consume flavonoid-rich fruits and vegetables benefit with an increased endothelium-dependent microvascular reactivity, the prevention of vascular stiffness, and reduced NO. The reduction of inflammatory biomarkers has also been observed [106,107] (see below). Although some studies suggest effects of fruit intake on AS (Table 3), it is clear that several factors limit the assessment of vascular and endothelial dysfunction in nutritional studies with fruit juice intake [108]. Heterogeneity in methodology and study design, limited data, bioavailability, and metabolism issues complicate interpretation [108].

Berry polyphenols (primarily flavonoids, isoflavonoids, anthocyanins, proanthocyanidins) have been investigated in a few small-scale studies for their beneficial effects on several surrogate markers of cardiovascular risk, including AS. Anthocyanins are probably the main bioactive compounds that characterize berries [109]. Although some promising activities can be noted for berries (see Table 3), the data are not sufficient to correlate berry polyphenol intake with improved AS. Bioavailability of polyphenols has been stressed as an important factor to be elucidated to allow a better understanding of this correlation [110]. Anti-inflammatory and antioxidant effects are frequently reported in an overview of berry consumption in metabolic syndrome patients [109].

In a review on the effect of polyphenols in grape juice (monomeric and oligomeric flavan-3-ols) on cardiovascular risk factors, a strong relationship between daily total polyphenol dose and change in FMD was observed. Also, more specifically, for Concord grape juice, clinically significant effects on FMD were found [111]. Grape extracts and wine seem to have some BP lowering effects, and a few reports point to an improved FMD, although conflicting reports have also been published. An effect on NO production and ET-1 synthesis has also been postulated [103] (Table 3).

### 8.3. Soy and Isoflavonoids

Evidence from trials on soy suggested a beneficial effect of soy (and isoflavones herein) intake on AS measured through PWV and arterial compliance and this could, at least in part, explain the low incidence of heart disease in populations with high soy intake [112]. This has been confirmed by Lilamand and coworkers, observing a decrease in PWV in healthy adults after isoflavone supplementation [15].

In spite of some diverging results, the importance of isoflavone metabolites in biological activities of polyphenols should not be neglected. This is demonstrated by the effect of *trans*-tetrahydrodaidzein, a metabolite typically formed after consumption of isoflavones (formononetin, daidzein), reducing BP and AS in obese men and postmenopausal women, in a double-blind, randomized and placebo-controlled trial with supplementation of *trans*-tetrahydrodaidzein [113]. Studies on soy and isoflavanoids are summarized in Table 4.

### 8.4. Miscellaneous Dietary Polyphenols

For olives and olive polyphenols, curcuminoids, walnuts, onion, lemon balm, and a composed polyphenol-rich beverage, isolated reports are listed in Table 5.

**Table 2 nutrients-11-00578-t002:** Cocoa, coffee, tea, and their isolated polyphenols in arterial stiffness.

Dietary Intervention/Polyphenol	Study Design	Health Status	Effects	References
Flavonoid-rich dark chocolate (*single dose of 100 g*)	17 young volunteers; randomized, single-blind, sham procedure-controlled, cross-over design	Healthy	↑ resting and hyperemic brachial artery diameter; ↑ FMD; ↓ Aix; No change in PWV	[114]
Cocoa (*0, 80, 200, 500, and 800 mg cocoa flavonoids/day/10 g cocoa in five periods of 1 week*)	20 volunteers; randomized, double-blind, controlled, cross-over design	Healthy	↑ FMD; ↓ PWV; ↓ BP; ↓ pulse pressure; ↓ ET-1	[115]
Flavanol-rich dark chocolate vs. flavanol-free white chocolate (*100 g/day for 3 days*)	12 volunteers	Healthy	Dark chocolate ingestion improved flow-mediated dilation (*p* = 0.03), wave reflections, endothelin-1 and 8-iso-PGF(2α) in contrast to white chocolate effects	[116]
Flavanol-rich dark chocolate vs. flavanol-free white chocolate, (*100 g/day for 15 days*)	19 volunteers (11 M); cross-over design	Hypertensive patients with IGT	↓ systolic and diastolic BP; ↑ FMD; ↑ insulin sensitivity;	[117]
Flavonoid-rich vs. flavonoid-poor dark chocolate	32 volunteers (16 M); sleep deprivation, randomized double-blind crossover design	Healthy	flavanol-rich chocolate promote: ↓ BP; ↓ pulse pressure; ↑ FMD; mitigated the increase in pulse-wave velocity	[118]
Cocoa flavanol-containing (***450 mg***) drink vs. cocoa flavanol-free control drink (*twice a day for 14 days*)	22 young (M) and 20 elderly (M) volunteers; randomized, controlled, double-masked, parallel-group dietary intervention trial	Healthy	↑ FMD in both groups; ↓ pulse wave velocity; ↓ total peripheral resistance-, ↑ arteriolar and microvascular vasodilator capacity; ↓ Aix in elderly	[119]
Cocoa beverage (*960 mg total polyphenols; 480 mg flavanols*)	18 volunteers; randomized, double-blind, crossover study	T2D	↓ large artery elasticity	[120]
Dark chocolate (***37 g/day***) and a sugar-free cocoa beverage (*total cocoa 22 g/day, total flavanols 814 mg/d, 4 weeks*)	30 middle-aged volunteers (15 M); randomized, placebo-controlled, cross-over study	Overweight	↑ basal diameter and peak diameter of the brachial artery and basal blood flow volume; ↓ Aix in only women	[121]
(2)-Epicatechin (***100 mg/d***), quercetin-3-glucoside (***160 mg/d***) or placebo (*capsules for 4 weeks*)	37 volunteers; a randomized, double-blind, placebo-controlled, crossover trial	Healthy	no effect on FMD, arterial stiffness	[122]
Dark chocolate (***70 g, 150 mg*** epicatechin) and pure epicatechin capsules (***2 × 50 mg*** epicatechin) with ***75 g*** white chocolate	20 (M) volunteers; randomized crossover study	Healthy	dark chocolate and epicatechin significantly ↑ FMD; ↓ Aix	[123]
Chlorogenic acid (***450 mg or 900 mg***) vs. ***200 mg*** (−)-epicatechin	16 volunteers; cross-over study	Healthy	no effect on BP; no significant effect on peak FMD response; ↑ post-ischemic FMD response	[124]
Coffee polyphenol extract (*355 mg chlorogenic acids*)	19 (M) volunteers; randomized, acute, crossover, intervention study	Healthy	↑ secretion of Glucagon-like peptide 1; ↑ postprandial hyperglycemia; ↑ FMD	[125]
Black tea (*0, 100, 200, 400, and 800 mg tea flavonoids/day in 5 periods of 1 week*)	19 (M) volunteers	Healthy	↑ FMD; ↓ blood pressure; ↓ stiffness index	[126]
Green tea (*9 g/day for 4 weeks*)	55 (31 M) volunteers; randomized, cross-over	T2D	No effect on brachial-ankle PWV; No effect on inflammatory markers	[127]
Green tea (*836 mg catechins, acute*)	20 volunteers; 2 different catechol-*O*-methyltransferase genotypes	Healthy	↓ digital volume pulse stiffness index (SI) in GG subjects; ↑ BP and insulin response in GG subjects	[128]

↑ increased; ↓ decreased; flow-mediated dilation (FMD); pulse wave velocity (PWV); endothelin-1 (ET-1); aortic augmentation index (Aix); plasma malondialdehyde (MDA); impaired glucose tolerance (IGT); Type 2 diabetes (T2D); male (M); blood pressure (BP); 8-isoprostane F2α (8-iso-PGF(2α).

**Table 3 nutrients-11-00578-t003:** Fruit, wine, and their isolated polyphenols in arterial stiffness.

Dietary Intervention	Study Design	Health Status	Effects	References
Apple with skin (*acute and 4 weeks*)	30 volunteers; randomized, controlled, cross-over	Healthy	↑ FMD	[129]
Red (anthocyanin-rich) or blond (anthocyanin-poor) orange juice (*1 liter, acute*)	18 volunteers (9 M); Randomized, cross-over design	Healthy	↓ Aix after red orange juice	[130]
Grapefruit juice (*340 mL/day* (*210 mg naringenin glycosides*)*, for 6 months*)	48 postmenopausal women; double-blind, randomized, controlled, cross-over	Healthy	↓ carotid-femoral PWV	[131]
Orange juice or hesperidin supplement (*acute intake; both 320 mg hesperidin*)	16 fasted volunteers (M)	Healthy	no effect on endothelial function; no effect on arterial stiffness; no effect on BP	[132]
Pomegranate extract-containing drink (*<50 mg pomegranate polyphenols per 237 mL*)	19 young volunteers (M); randomized, controlled, crossover	Healthy	no effect on digital volume pulse-stiffness index	[133]
Pomegranate juice (*330 mL/day for 4 weeks*)	51 adults volunteers (16 M)	Healthy	no effect on PWV; ↓ systolic and diastolic BP; ↓ mean arterial pressure	[134]
Mango fruit preparation Careless™ (*single dose of 100 mg or 300 mg*)	10 volunteers (F); randomized, double-blind, crossover pilot study	Healthy	↑ coetaneous blood flow; No effect on endothelial function	[135]
Cranberry juice cocktail (*500 mL/day* (*27% juice*) *for 4 weeks*)	35 volunteers (M); double-blind, cross-over	Healthy	no effect on Aix; ↓ in Aix significant within-group in abdominally obese M	[136]
Cranberry juice (*54% juice, 835 mg total polyphenols, and 94 mg anthocyanins, for 4 weeks*)	15 volunteers; acute pilot study 44 volunteers; chronic placebo-controlled crossover	Coronary heart disease	↑ brachial artery FMD and digital pulse amplitude tonometry ratio in the pilot study; ↓ carotid-femoral PWV for chronic treatment	[137]
Blueberry (*300 g of blueberry*)	16 smokers (M) 3-armed randomized-controlled	Healthy	↓ peripheral arterial dysfunction; no differences in digital augmentation index	[138,139]
Blueberry (*300 g of blueberry*)	24 volunteers (M) (12 non-smokers and 12 smokers)	Healthy	↓ peripheral arterial dysfunction (reactive hyperemia index); no change in digital augmentation index dAix	[140]
Blueberry powder (*22 g freeze-dried, for 8 weeks*)	48 postmenopausal (F); randomized, double-blind, placebo-controlled	Pre- and stage 1-hypertension	↓ systolic and diastolic BP; ↓ brachial-ankle PWV	[141]
Strawberry powder (*40 g freeze-dried*)	30 overweight or obese adults (17 M)	Healthy	no effect on vascular function	[142]
Blackcurrant extract (*low sugar fruit drinks containing 150, 300, and 600 mg of total anthocyanins, acute*)	14 (M) and 9 postmenopausal (F); randomized, double-blind, cross-over	Healthy	no effect on arterial stiffness; no effect on 8-isoprostane F2α	[143]
Black raspberry (*750 mg/day, acute, and 12 weeks*)	26 and 39 volunteers, respectively	Metabolic syndrome	↓ augmentation index acutely; ↑ brachial artery FMD after 12 weeks of treatment	[144,145]
Concord grape juice (*7 mL/kg/day, 70-kg person consumed 490 mL/day; 965 mg total polyphenols and 327 kcal, 2-weeks*)	26 healthy smokers (10 M); randomized, placebo-controlled, double-blind, cross-over	Healthy	↑ values of FMD and PWV	[146]
Grape seed extract (*150 mg twice daily, 6 weeks*)	29 middle-aged (15 M); single-center, randomized, two-arm, double-blinded, placebo-controlled	Pre- hypertension	↓ systolic and diastolic BP; no significant changes in FMD	[147]
Grape-wine extract (*capsules MegaNatural™ combined with Provinols™, 4 weeks*)	60 volunteers; double-blind, placebo-controlled, crossover	Mildly Hypertensive, untreated	↓ 24-h ambulatory systolic and diastolic BPs; no effect on FMD	[148]
Red wine (*400 mL, ~13%* (*v/v*) *alcohol, 6 weeks*)	45 postmenopausal women; randomized parallel-arm	Hypercholesterolemia	↓ Aix; no effect on central hemodynamic parameters	[149]
Resveratrol (*100 mg tablet, oligo-stilbene 27.97 mg/100 mg/day, 12 weeks*)	25 volunteers (15 M); double-blind, randomized, placebo-controlled	T2D	↓ systolic BP; ↓ cardio-ankle vascular index	[150]
Resveratrol (*resVida™; 6 capsules, 30, 90, and 270 mg, single dose*)	19 volunteers (14 M); double-blind, placebo-controlled	Overweight/obese/post-menopausal untreated borderline hypertension	↑ FMD response	[151]
Resveratrol (*Resvida, 75 mg capsule/day, 6 weeks*)	28 obese volunteers (12M); randomized, double-blind, placebo-controlled crossover	Healthy	↑ FMD response; no effect on BP and arterial compliance	[152]

↑ increased; ↓ decreased; flow-mediated dilation (FMD); aortic augmentation index (Aix); pulse wave velocity (PWV); blood pressure (BP); Type 2 diabetes (T2D); male (M); female (F).

**Table 4 nutrients-11-00578-t004:** Soy and isoflavonoids in arterial stiffness.

Dietary Intervention	Study Design	Health Status	Effects	References
Isoflavone, red clover-extracted (*500-mg tablets, 2 × 40 mg of isoflavones/day, 6 weeks*)	80 volunteers (46 M); randomized, double-blind, cross-over, placebo-controlled	Healthy	improved arterial stiffness; ↑ systemic arterial compliance; ↓ total peripheral resistance; ↓ central PWV	[153]
Isoflavone (*50 mg/day, as black soybean tea, 2 months*)	55 volunteers (F); smokers and nonsmokers	Healthy	↓ cardio-ankle vascular index in premenopausal; no effect in postmenopausal; no effect on BP and brachial-ankle PWV	[154]
Isoflavone-containing soya protein isolate (*50 g/d soya protein, 6 weeks*)	20 volunteers (9 M); randomized, placebo-controlled, cross-over	Moderately elevated brachial BP	↓ brachial diastolic BP; no effect on Aix and PWV	[155]
Flavonoid-enriched chocolate (*split dose of 27 g/day* (*850 mg flavan-3-ols* (*90 mg epicatechin*)) *+ 100 mg isoflavones* (*aglycone equivalents*)*/day, 1 year*)	93 postmenopausal volunteers; double-blind, parallel-design, placebo-controlled	T2D	no change in intima-media thickness of the common carotid artery Aix or BP improved pulse pressure variability; equol producers had larger ↓ in diastolic BP, mean arterial pressure, and PWV	[156]
Soy germ pasta (*80-g serving/day, naturally enriched in isoflavone aglycons, 4 weeks*)	62 volunteers (25 M); randomized, controlled, parallel study	Hypercholesterolemia	improved arterial stiffness; the best effect in equol producers	[157]
Soy germ pasta (*one serving/day of* (*31–33 mg) total isoflavones*)*, 8 weeks*)	26 volunteers (13 M); randomized, controlled, double-blind, crossover	T2D	improved arterial stiffness; ↓ systolic and diastolic BP	[158]
Isoflavone capsule (*80 mg aglycone equivalents of daidzein and genistein, a SoyLife extract* (*40%*) *with a typical soy germ ratio of genistein:daidzein:* *glycitein* (*15:50:35*)*, acute*)	28 volunteers; equol producer phenotype (14 M), double-blind, placebo-controlled crossover	Healthy	improved carotid-femoral PWV in equol producers; no vascular effects	[54]
Soy nuts snack (*70 g of soy nuts: 101 mg of aglycone equivalents* (*55 mg of genistein, 42 mg of daidzein, and 4 mg of glycitein*)*, 4 weeks*)	17 volunteers (12 postmenopausal F, 5 M)	Metabolic syndrome	improved arterial stiffness (Aix)	[159]

↑ increased; ↓ decreased; aortic augmentation index (Aix); pulse wave velocity (PWV); blood pressure (BP); Type 2 diabetes (T2D); male (M); female (F).

**Table 5 nutrients-11-00578-t005:** Miscellaneous dietary polyphenols in arterial stiffness.

Dietary Intervention	Study Design	Health Status	Effects	References
Olive leaf extract (*51 mg oleuropein; 10 mg hydroxytyrosol, acute*)	18 volunteers (9 M); randomized, double-blind, placebo-controlled, cross-over	Healthy	↓ digital volume pulse-stiffness index; ↓ *ex vivo* IL-8 production	[160]
Red yeast rice and olive fruit extract (*9, 32 mg hydroxytyrosol*)	50 volunteers; randomized, double-blind, placebo-controlled	Metabolic syndrome	↓ SBP and DBP; ↓ LDL and oxidized LDL; ↓ lipoprotein-associated phospholipase A_2_	[161,162]
Olive fruit extract (*50 mg and 100 mg hydroxytytosol*)	36 volunteers; 11-day, double-blind, placebo-controlled	Risk for arterial stiffness	↓ Cardio-Ankle Vascular Index	[163]
Polyphenols (*250 mL beverage: 361 mg of* (*poly*)*phenols, 120 mg vitamin C; twice/day, 4 weeks*)	20 volunteers (10 M); a randomized, double-blind, placebo-controlled design	Healthy	No effect on the cutaneous vascular response; No effect on PWV	[164]
Curcumin capsules (*250 mg of curcuminoids, 3 capsules/twice a day, 6 months*)	107 volunteers (50 M); randomized, double-blinded, placebo-controlled	T2D	↓ PWV	[165]
Curcumin (*25 mg of highly absorptive curcumin dispersed with colloidal nanoparticles, 6 pills/day, 8 weeks*)	32 sedentary postmenopausal women (F)	Healthy	↓ FMD	[166]
Walnut-enriched ad libitum diet (*56 g of shelled, unroasted English walnuts/day, 8 weeks*)	46 volunteers (18 M); randomized, controlled, single-blind, crossover clinical	Overweight	↓ FMD, beneficial trends in systolic BP reduction	[167]
Onion skin extract (*162 mg/day quercetin, 6 weeks*)	70 volunteers (35 M); double-blinded, placebo-controlled cross-over	Healthy	↓ 24 h systolic BP in the subgroup of hypertensives	[168]
Lemon balm extract (*3.3 g of lemon balm leaves extracted in 200 mL of hot water, once daily, 4 weeks*)	28 Japanese volunteers (14 M); an open-label, parallel-group comparative	Healthy	↓ in brachial-ankle PWV	[169]

↑ increased; ↓ decreased; Flow-mediated dilation (FMD); pulse wave velocity (PWV); blood pressure (BP); systolic blood pressure (SBP); diastolic blood pressure (DBP); interleukin 8 (IL-8); low-density lipoprotein (LDL); Type 2 diabetes (T2D); male (M); female (F).

## 9. Impact on Mechanisms Contributing to AS

Vascular, antioxidant, anti-inflammatory, antiglycation, and autophagy inducing effects are the most important mechanisms contributing to the pathophysiology of AS, that are addressed to evaluate the impact of dietary phenolics. However, several of these mechanisms, such as oxidative stress and inflammation, are closely interrelated, as oxidative stress can cause inflammation, which, in turn, can induce oxidative stress. Both oxidative stress and inflammation cause injury to endothelial cells. Endothelial dysfunction consecutively promotes a pro-inflammatory environment, resulting, among others, in an increased expression of adhesion molecules. As a positive feedback loop, vascular inflammation leads to endothelial dysfunction [170]. The antioxidant properties of polyphenols can also contribute to antiglycation. Furthermore, transcription factors and signaling pathways involved in oxidation and inflammation have been implicated in autophagy.

### 9.1. Vascular Effects

Endothelial function is regulated by NO (vasodilating) and endothelin (vasoconstricting). Human trials have demonstrated vasoprotective effects mediated by NO, which is produced by eNOS. Adequate production and bioavailability of eNOS-derived NO are necessary for the maintenance of a healthy endothelium; reduced eNOS-derived NO bioavailability results in endothelial dysfunction [171]. On the other hand, blood pressure relies on the renin-angiotensin-aldosterone system (RAAS), producing the vasoconstrictive Ang II from angiotensin I (Ang I) by the angiotensin-converting enzyme (ACE).

Flavonoids display antihypertensive effects by increasing NO production in endothelial cells, as well as by direct inhibition of ACE [15]. Several dietary polyphenols increase the production or bioavailability of endothelial NO, as seen for cocoa flavanols. Their vasodilatory response is NO-dependent and can be reversed by blocking nitric oxide synthesis, as reported in vitro as well as in human trials [171].

ET-1, on the other hand, is a potent vasoconstrictor peptide with pro-oxidant and pro-inflammatory properties and plays a role in the development of endothelial dysfunction. ET-1 expression and production in endothelial cells are, among others, increased by Ang II-stimulation and aging. ET-1 overexpression activates NADPH oxidase, and therefore ROS formation, causing oxidative stress and forming a positive feedback loop of oxidative stress-mediated endothelial oxidative injury and dysfunction. Moreover, oxidative stress also causes amplification of the ACE activity, subsequently stimulating the angiotensin II receptor type 1 (AT-1 receptor) by Ang II, and thus inducing the production of ROS by NADPH oxidase and amplifying the detrimental process [170].

Several acute and short-term trials have investigated the effects of flavonoid-rich foods and beverages on FMD as a marker of endothelial function, reporting an increase in FMD of about 20–30% [172]. Phytoestrogens (isoflavonoids genistein, daidzein), flavonoids (e.g., artemetin), anthocyanins (e.g., delphinidin) induce vasodilation by binding to estrogen receptors in physiologically relevant concentrations, leading to increased eNOS activity and increased NO synthesis [37]. Inhibition of ET-1 release in human umbilical vein endothelial cells (HUVECs), together with increased eNOS expression has been demonstrated for delphinidin and cyanidin. For delphinidin glycosides, as for other polyphenols, stimulation of endothelin B receptors and inhibition of ACE have also been reported in vitro [37].

The specific mechanisms, by which cocoa flavanols improve vascular function are still under investigation but could be linked to the modulation of NADPH oxidase, to maintain low levels of superoxide radical anion not affecting the vascular endothelium. Increased activity of NADPH oxidase is implicated in vascular dysfunction [171]. Cocoa supplementation has been shown to decrease both SBP and DBP in several studies. In a 2014 review, Latham et al. concluded that cocoa flavanols’ beneficial cardiovascular effects are the result of increased NO bioavailability [173,174]. Antioxidant activity can contribute to an enhanced endothelial function. NO degradation is modulated by free radicals, and therefore vascular function is also affected by antioxidant (and anti-inflammatory) actions, as observed for cocoa [33].

Cocoa intake for 4 weeks significantly decreased postprandial SBP in obese subjects, independently of body weight loss, while bioavailability of cocoa components was confirmed by the analysis of 14 derived metabolites in plasma [175]. Blood pressure reduction by cocoa through stimulation of eNOS activity was also observed by Ludovici and coworkers, in addition to an increase in l-arginine bioavailability caused by reduced arginase activity, inhibition of ET-1 production and of l-NAME [33]. Additionally, ACE-inhibition by procyanidin-rich chocolate has also been observed [33,174]. Oligomeric procyanidins are reported to stimulate endothelium-dependent vasodilation, suppress ET-1 synthesis, and inhibit the activity of ACE, resulting in blood-pressure-lowering effects. However, the bioavailability of those oligomers is an important factor for translation into in vivo effects [176].

A similar effect has also been observed for tea flavanols (−)-epicatechin, (−)-epigallocatechin, and their gallates whose ACE inhibition in HUVEC cells was dose-dependent [174,177]. Interference of flavonoids in blood pressure regulation by RAAS results in a lower production of superoxide anion by NADPH oxidase [174].

Resveratrol has been shown to increase endothelial NO production, thereby improving endothelial dysfunction and lowering BP in hypertensive rats, which is explained by calcium-dependent eNOS activation [178]. Morin, a flavonol present in the *Moraceae* family, protects against endothelial dysfunction through an Akt (protein kinase B) -dependent activation of eNOS signaling in a diabetic mouse model [179].

Adenosine monophosphate-activated protein kinase (AMPK) is an important sensor of cell energy status and can be activated by stressors, such as oxidative stress, hypoxia, and nutrient deprivation. Targets of AMPK include enzymes of glucose and lipid metabolism, mitochondrial enzymes, and eNOS [172]. EGCG is able to increase cytosolic calcium concentrations, contributing to NO production by binding to calmodulin in the heart and vascular endothelium. Furthermore, it activates AMPK and, consequently, reduces ET-1 expression [180].

Resveratrol ingestion in mice stimulates the activities of sirtuin 1 (SIRT1) and AMPK, both of which influence the regulation of metabolism [181]. Resveratrol and related stilbenoids pterostilbene and gnetol attenuate the increase in media-to-lumen ratio and wall component stiffness observed in a rat model of spontaneously hypertensive heart failure. However, the authors could not demonstrate a role of AMPK or extracellular signal-regulated kinases (ERK) herein [182]. In contrast, an attenuation of the vascular geometry remodeling process and ERK-signaling by resveratrol, rather than a direct effect on arterial wall stiffness, has been observed in spontaneously hypertensive rats [183]. Also, in spontaneously hypertensive rats, whole grape extract promoted a tendency to reduce arterial wall component stiffness, although not significantly. Reduced blood pressure and improved vascular function and compliance were, however, observed, which were not only due to the grape resveratrol but rather to other grape components [184]. A low-molecular grape seed polyphenol extract, rich in flavanols, decreased plasma ET-1, up-regulating eNOS and SIRT-1 and down-regulating aortic gene expression of ET-1 and NADPH in rats, indicating the vasoprotective effect of grape seed flavanols [185].

Curcumin supplementation in young and old mice resulted in amelioration of age-associated large elastic artery stiffening (PWV), NO-mediated vascular endothelial dysfunction, oxidative stress, and increase in collagen and AGEs [186].

Urolithins, gut-derived ellagitannin metabolites, activated eNOS in human aortic endothelial cells [187]. In the case of the daidzein metabolite equol, it activates eNOS via Akt and extracellular signal-regulated kinase 1/2-dependent signaling and mitochondrial superoxide generation [188]. In line with this, in both porcine pulmonary arteries and in human pulmonary artery endothelial cells, equol reversed ritonavir-induced endothelial dysfunction, including a reduction in the vasomotor dysfunction, eNOS downregulation, and oxidative stress [189]. In vivo, refeeding of isoflavones to rats on an isoflavone-deficient diet led to increased production of nitric oxide and endothelium-derived hyperpolarizing factor (EDHF), up-regulation of antioxidant defense enzymes, and lowering of blood pressure [190].

### 9.2. Oxidant Status

The chemical structure of several polyphenols is ideal for scavenging of free radicals and reactive oxygen species. The aromatic feature and highly conjugated system with multiple hydroxyl groups make these compounds excellent electron or hydrogen atom donors, neutralizing free radicals and other ROS. Therefore, dietary phenolics are powerful antioxidants in vitro [38]. Differences exist depending on the number and location of the free hydroxyl groups—especially the presence of catechol groups is important—and on the electron deficiency in anthocyanins. Besides direct scavenging of ROS and reactive nitrogen species (RNS), polyphenols can react with the peroxidation products of macromolecules, such as lipids, proteins, DNA, and RNA, or act as metal chelators.

Nevertheless, the promising in vitro antioxidant capacity cannot easily be extrapolated to an in vivo situation due to the limited bioavailability of polyphenols [37]. The plasma concentration of flavonoids is typically insufficient (less than 1 μmol/L) to exert significant antioxidant activities via direct radical scavenging or reducing power, measurable by the existing in vitro assay methods. The complex intrinsic antioxidant system also makes it difficult to validate the systemic antioxidant effects of the poorly absorbed phenolic compounds in vivo [38].

Although polyphenols have been linked to a reduced risk for CVD, primarily indicated by altered biomarkers of oxidative stress, a causal link is more difficult to prove [37]. Moreover, high consumption of antioxidant polyphenols has a noxious pro-oxidant effect, thus stimulating oxidation of biomolecules. On the other hand, moderate pro-oxidant effects could also turn out to be beneficial, by stimulation of the intracellular antioxidant defense mechanisms, such as antioxidant enzymes [37]. Indeed, it has become clear that the antioxidant effect goes beyond direct interference with ROS [191]. Modulation of ROS production in mitochondria, NADPH oxidases, and uncoupled eNOS, together with up-regulation of antioxidant enzymes, such as glutathione-*S*-transferase (GST), SOD, glutathione reductase (GR), quinone oxidoreductase 1, Hmox-1, and glutamyl-cysteine ligase (GSL), is more relevant [192]. Hmox-1 is a key regulator of endothelial function and is involved in vascular protection against ROS-induced oxidative damage, and AMPK activation results in its expression through the nuclear factor (erythroid-derived 2) (NFE2) - related factor 2 (Nrf2)/antioxidant response element protein (ARE) pathway [193].

A proposed mechanism for ‘nutritional antioxidants’, like polyphenols, involves the paradoxical oxidative activation of the Nrf2 signaling pathway (Figure 4). Nrf2 can be activated by ROS in the cytoplasm, after which it is translocated to the nucleus and regulates ARE-mediated transcriptions of various genes encoding the above-mentioned antioxidant enzymes [37,38]. Nrf2 is under constant control of the redox-sensitive repressor protein Keap1 (Kelch-like ECH-associated protein 1) [194].

Low concentrations of phenolic compounds (or their metabolic products) and the quinones formed under the influence of interactions with ROS are electrophiles that can interact with Keap1, and thus lead to activation of the redox-sensitive Nrf2 [172,195]. The consumption or supplementation of dietary polyphenols has indeed been shown to restore redox homeostasis inducing an antioxidant response in target cells using the Nrf2/ARE pathway thus inducing detoxifying enzymes.

Resveratrol, for instance, demonstrates a wide range of biological effects, of which many are related to its ability to activate the Keap1/Nrf2/ARE signaling system. Additionally, it inhibits transcription factors NF-κB, activator protein 1 (AP-1), p53 and activates kinases MAPK, Akt, AMPK, phosphoinositide 3-kinase (PI3K), as well as SIRT1. Also, flavonoids, including catechins like EGCG and hydroxytyrosol from olives, have been shown to induce this antioxidant signaling system [194]. Modulation of such antioxidant signaling cascades by polyphenols recently has been recently evidenced extensively in in vitro and animal models [37]. Flavonoids modulate different signaling cascades, such as PI3K, Akt/PKB, tyrosine kinases, protein kinase C (PKC), and MAPK [174]. In addition, flavonoids also modulate the expression of various genes through activation of a broad range of transcription factors [97]. Curcumin activates the Keap1/Nrf2/ARE system and induces the expression of antioxidant genes [194]. The vascular protection and antioxidant effects of soy isoflavone diets are attributed mostly to an up-regulation of eNOS expression and activity, increased NO bioavailability associated with Nrf2 accumulation, and ARE dependent activation of antioxidant defense enzymes by both the isoflavones and their microbial metabolites, including equol [102,196,197,198] (Figure 4). Cocoa flavanols can directly interact with ROS but exhibit antioxidant effects indirectly through modulation of crucial oxidative stress-related enzymes: induction of antioxidant enzymes and inhibition of pro-oxidant enzymes like NADPH oxidase [199]. Anthocyanin-rich beverages increased SOD and catalase, and decreased malondialdehyde, a biomarker of lipid peroxidation, without affecting inflammatory biomarkers in healthy women [200]. Chlorogenic acid has been shown to protect against hypochlorous acid (HOCl)-induced oxidative damage in mouse endothelial cells *ex vivo*, via increased production of NO and induction of Hmox-1 [193]. Also, enterolactone induced Hmox-1 expression through Nrf2 activation in endothelial cells [201]. Moreover, flavonols and isoflavones can regulate aryl hydrocarbon receptor (AhR)-mediated signaling in cells, thus influencing Nrf-2 translocation [38].

Additionally, dietary polyphenols can also suppress oxidative stress by interfering with inflammatory signaling cascades controlled by NF-κB and MAPK. Activation of these cellular processes leads to induction of regulatory immune responses. As a result, pro-inflammatory cytokines, including interleukin (IL)-1β, IL-6, IL-8, tumor necrosis factor (TNF)-α, and interferon (IFN)-γ, are released [38].

### 9.3. Anti-Inflammatory Activity

Antioxidant and anti-inflammatory pathways influenced by dietary polyphenols are largely intertwined and can affect similar biomarkers [38]. NF-κB is a central transcription factor in inflammation (Figure 5), which stimulates the encoding of several genes, including those responsible for producing cytokines, chemokines, immunoreceptors, cell adhesion molecules, and acute-phase proteins. The activation of NF-κB is redox-sensitive and direct inhibition of NF-κB by polyphenols (e.g., resveratrol and curcumin analogs) is an important mechanism for their anti-inflammatory effects [37].

Nucleotide-binding oligomerization domain, leucine-rich repeat containing gene family, and pyrin-domain containing 3 (NLRP3) inflammasome is a key node that links the signaling cascades between the antioxidant response and inflammation and has recently been shown to be modulated by polyphenols. Increased ROS activates NLRP3, which induces IL-1β, and via the toll-like receptor (TLR)-1, it triggers NF-κB-activated and MAPK-induced pro-inflammatory signaling, producing inflammatory cytokines, such as IL-1β, IL-6, IL-8, TNF-α, and IFN-γ [38]. Several reports on the modulation of NLRP3 activation by polyphenols in cell systems and rat liver tissue (e.g., resveratrol, procyanidin B2, chlorogenic acid) resulting in an anti-inflammatory effect, have been published recently [202,203,204].

Anthocyanins also target the MAPK pathway [37]. Polyphenols, including flavonoids, have also been reported to stimulate peroxisome proliferator-activated receptor γ (PPAR-γ) or SIRT-1 mediated signaling, and to interfere with TNF-α-induced MAPKs and NF-κB pro-inflammatory signaling transductions, resulting in the repression of inflammation [38]. In a small study in insulin deficient and insulin resistant diabetic rat models, baicalein ameliorated blood pressure elevations and exhibited both antiglycation (AGEs) and anti-inflammatory (NF-κB, TNF-α) mechanisms [205].

Curcumin is a potent multi-targeted polyphenol modulating multiple cell signaling pathways linked to different chronic diseases. It has been shown to exhibit anti-inflammatory effects by down-regulating various cytokines, such as TNF-α, IL-1, IL-6, IL-8, IL-12, monocyte chemoattractant protein (MCP)-1, and IL- 1β, and various inflammatory enzymes and transcription factors [206].

Dietary polyphenols can also modulate pro-inflammatory NF-κB signaling through targeting the RelB/AhR complex, which is also involved in redox management due to binding with the xenobiotic responsive element. Dietary flavonoids are known to act as AhR modulators [38].

Cell adhesion molecules, including intercellular adhesion molecule-1 (ICAM-1), vascular cell adhesion molecule-1 (VCAM-1), and endothelial selectin (E-selectin), are glycoproteins involved in tissue integrity, cellular communication and interactions, and extracellular matrix contact. They are increased in endothelial dysfunction, vascular remodeling, and obesity [25]. Polyphenols also display anti-inflammatory properties by inhibiting adhesion molecule (VCAM, ICAM-1) production by the endothelium. In vitro, fourteen phenolic acid metabolites and six flavonoids were screened for their relative effects on VCAM-1 secretion by HUVECs stimulated with TNF-α. Protocatechuic acid was the most active of the phenolic metabolites, while native flavonoids showed no activity in HUVEC cells [207]. ICAM-1 expression was reduced in endothelial cells by urolithin A [208]. In vivo, however, conflicting results exist on the modulation of cell adhesion molecules (VCAM and/or ICAM) after interventions with polyphenol-containing foods [192]. Resveratrol ameliorated aortic stiffness (PWV) in mice with metabolic syndrome by activation of vascular smooth muscle sirtuin-1, associated with a decrease in NF-κB activation and VCAM-1 [209]. VCAM-1, NO, and malondialdehyde concentrations were lower in equol producing compared with non-producing post-menopausal women after supplementation with soy isoflavones [210]. However, the difference was only significant for malondialdehyde.

Expression of other pro-inflammatory mediators, such as cyclooxygenase 2 (COX-2), is suppressed by various flavonoids and anthocyanins [37], for example, the beneficial effect of flavonoids on intestinal inflammation has directly been related to the suppression of pro-inflammatory enzyme expression, such as COX-2 and iNOS [36]. Polyphenols from red wine and black tea (quercetin, EGCG, epicatechin gallate, and theaflavins) are able to inhibit COX-2 and lipoxygenase in a dose-dependent manner in lipopolysaccharide (LPS)-activated murine macrophages [102]. LPS-induced inflammation was attenuated via (among others) suppression of COX-2 expression also by phenolic metabolites, such as urolithins or equol, in various models [211,212].

CRP and high-sensitivity CRP values were associated with AS in patients with metabolic syndrome, renal transplant, diabetes mellitus, and rheumatoid arthritis [25]. Resveratrol significantly reduced CRP while increasing Total Antioxidant Status values in smokers [213]. Resveratrol was tested in various studies, sometimes combined with grape extract, and generally showed a decrease in CRP, IL-1β, ICAM, TNF-α expression, and IL-10 in different populations [214]. Red wine phenolics decreased serum levels of ICAM-1, E-selectin, and IL-6 in a randomized cross-over trial on male volunteers [215].

In a 2011 meta-analysis, Dong and coworkers concluded that there was insufficient evidence for the significant reduction of CRP by soy isoflavones in postmenopausal women, in general. However, soy isoflavones may reduce CRP significantly among postmenopausal women with elevated CRP [216]. Several other inflammatory biomarkers have been investigated after isoflavone consumption, but without consistent conclusions [214]. Urinary excretion of daidzein and its metabolites *O*-desmethylangolensin and equol were negatively inversely associated with serum CRP in an analysis involving 1683 participants [217].

Lignans, however, are not effective in reducing CRP in the general population but do show a significant decrease in obese subjects [218]. Urinary total and individual phytoestrogens, including lignans and enterolactone, were significantly inversely associated with serum CRP in a nationally representative sample of 6009 subjects from the U.S. High urinary enterolactone, but not enterodiol concentration was found to be inversely associated with obesity, abdominal obesity, high serum CRP, high serum triglycerides, low serum high-density lipoprotein (HDL) cholesterol, and metabolic syndrome in an analysis including 21,776 subjects [219,220].

Cocoa powder and epicatechin have been shown to decrease several inflammatory markers, such as TNF-α, IL-6, IL-10, and also CRP. Flavonoid-rich fruit and vegetable intake reduced CRP, E-selectin, and VCAM in men with increased CVD risk [106]. In contrast, pomegranate juice did not exhibit a significant effect on CRP levels in a meta-analysis of five prospective trials [221].

Olive oil polyphenols are able to decrease inflammatory markers, such as thromboxanes, leukotrienes, cytokines, CRP, and soluble adhesion molecules, in humans [222]. A traditional Mediterranean diet, including polyphenol-rich virgin olive oil, decreased plasma oxidative and inflammatory status and the corresponding gene expression in peripheral blood mononuclear cells of healthy volunteers, indicating that the benefits associated with a Mediterranean diet and olive oil polyphenol consumption on cardiovascular risk can be mediated, at least in part, through nutrigenomic effects [223]. Catechins and curcumin, for instance, are found to regulate MMPs expression in diverse models, including VSMC [102]. Also, red grape skin extracts and their polyphenols (trans-resveratrol, trans-piceid, kaempferol, and quercetin) inhibit endothelial invasion, as well as the MMP-9 and MMP-2 (gelatinases) release in stimulated endothelial cells and MMP-9 production in monocytes, at concentrations likely to be achieved after moderate red grape skin consumption [224].

Moreover, in view of the anti-inflammatory activity of polyphenols (and/or their metabolites), an effect on chronic inflammation is very likely. This is also discussed in a recent review, which reports multiple inflammatory components targeted by polyphenols, thus leading to anti-inflammatory properties. The effects of polyphenols on the immune system are associated with extended health benefits for different chronic inflammatory diseases. Studies of plant extracts and compounds show that polyphenols can play a beneficial role in the prevention and the progress of chronic diseases related to inflammation, such as diabetes, obesity, neurodegeneration, cancers, and cardiovascular diseases, among other conditions [225].

### 9.4. Antiglycation/AGEs

AGEs are the end products of the Maillard reaction, in which proteins or amino acids react with reducing sugars. The crosslinking of vascular collagen by AGEs increases AS. Soluble receptors for AGE (sRAGE) levels are inversely correlated with the urinary excretion of isoprostanes (8-iso-PGF2α), a biomarker of lipid peroxidation, suggesting a link between vascular stiffness and oxidative stress [22].

Polyphenols can exhibit antiglycation function through their influence on glucose metabolism (aldose reductase inhibition), antioxidant properties, protein and receptor binding, modulation of gene expression and dicarbonyl (e.g., methylglyoxal = a highly reactive intermediate) trapping properties [226]. A bulk of evidence exists for the in vitro antiglycation properties of polyphenols. Some polyphenols have been shown to be even more effective than the reference compound aminoguanidine in inhibiting glycation in vitro [12]. Xie and Chen reviewed in vitro and animal studies on the anti-glycation activity of polyphenols, to extract a structure-activity relationship [227]. Lemon balm (Melissa officinalis) has been selected as the most active extract from 681 hot water extracts in a pentosidine formation assay, with a higher potency than aminoguanidine. Rosmarinic acid has been identified as the major active compound herein [169].

Redox-active transition metals, such as Cu ^2+^, Fe ^2+^, Mn ^2+^, and Zn ^2+^, catalyze carbonyl formation in proteins. Inhibitors of AGE formation, which have been successful at decreasing arterial stiffness, have known ability to chelate metals [11]. Pomegranate (*Punica granatum*) extract and its polyphenolic constituents punicalin, punicagalin, ellagic, and gallic acid significantly suppressed AGE formation in vitro and in a mouse model [228]. Also, rambutan (*Nephelium lappaceum*) extract exhibited antiglycation activity in vitro, which correlated with its antioxidant activity. Main compounds were geraniin and ellagic acid [229]. Similarly, glucitol-core containing gallotannins isolated from maple (*Acer*) species inhibited AGEs, mediated by their antioxidant (radical trapping) potential [230]. For quercetin, researchers demonstrated that the antiglycation activity in vitro was due to its metal chelating, methylglyoxal trapping, and ROS trapping properties [231].

In a recent review on polyphenols with antiglycation activity, Yeh et al. reported on the antiglycation potential of different polyphenol classes. The number of –OH groups seems important for the activity. Simultaneous use of multiple polyphenol types could add to their efficacy [232]. This is also illustrated by the in vitro antiglycation activity of an olive leaf extract and two characterized fractions. Both the inhibition of early and advanced stage glycation was observed. However, each fraction separately was not able to show the same activity, indicating that compounds from both fractions are necessary for the effect. Hydroxytyrosol in synergy with minor compounds with similar polarity seemed responsible for the antiglycation activity in a hepatic cell line [233,234]. More in vivo studies should clarify the relevance of dietary polyphenols in protection against AGE-depending conditions like arterial stiffening.

### 9.5. Autophagy

Natural compounds with anti-inflammatory and antioxidant activity, such as polyphenols, are potentially useful to prevent arterial stiffness by promoting autophagy. Mammalian target of rapamycin (mTOR) is a central protein kinase that suppresses autophagy and is under control of kinase signaling cascades, including the autophagy activating AMPK pathway [194].

Limited results are available for resveratrol, activating SIRT-1 and AMPK in endothelial and smooth muscle cells *in vitro,* inducing smooth muscle cell differentiation and thus maintaining vascular plasticity [27]. Also, in rhesus monkeys, resveratrol, which has been shown to activate endothelial autophagy, reduced arterial aging [18]. Curcumin and analogs are effective stimulators of autophagy, by modulating several transcription factors (NF-κB, Nrf2, AP-1, HIF-1). Green tea polyphenols alleviated autophagy inhibition induced by high glucose in endothelial cells. Autophagy modulation may be involved in the endothelial protective effects of green tea against hyperglycemia [235]. EGCG was able to enhance autophagy-dependent survival through modulation of mTOR-AMPK pathways in a human embryonic kidney cell line [236]. However, modulation of autophagy by EGCG seemed to be dependent on cell type, stress condition, and concentration: low levels of EGCG increased autophagy, while higher levels decreased this process [180]. EGCG increased the formation of LC3-II and autophagosomes, and therefore stimulated autophagy in bovine endothelial cells [237].

Hydroxytyrosol was able to induce autophagy, and this was associated with a lower inflammatory response in vascular adventitial fibroblasts [238]. The activation of the SIRT1 signaling pathway is thought to play an important role in this process [238,239,240]. Hydroxytyrosol has been reported to induce autophagosome formation, but at the same time to inhibit their degradation by lysosomes in cancer cells; reports in endothelial cells or VSMC are currently lacking [194].

Additionally, oleuropein (OLE)-aglycone has been shown to induce autophagy through AMPK/mTOR-mediated signaling in neuroblastoma cells and an OLE-fed mouse model of amyloid beta (Aβ) deposition [241]. An increased autophagic flux was found to contribute to the anti-inflammatory potential of urolithin A, associated with impaired Akt/mTOR (mammalian target of rapamycin) signaling in murine J774.1 macrophages [242].

## 10. Conclusions

AS is an important risk factor for cardiovascular morbidity and mortality and is reflected by structural and functional changes in the vessel wall. It has received considerable interest as a relevant target for patients with increased cardiovascular risk. Nutrition and food-related compounds can offer a suitable strategy in the prevention or reversal of AS.

Considering the different mechanisms involved in the pathophysiology of AS, dietary polyphenols offer an array of relevant activities, interfering with vascular, oxidative, inflammatory, glycation, and autophagy pathways, and could therefore potentially be effective at counteracting or preventing age-induced vascular stiffening. The elucidation of the exact mechanisms of action and targets for native polyphenols, and more importantly for their metabolites, has largely been neglected and requires further studies. In vivo beneficial effects of polyphenols on AS, and therefore protective effects against cardiovascular risk, have been reported, although often findings have been inconclusive or even inconsistent. Most evidence exists on cocoa and flavanols therein, isoflavones and anthocyanins.

However, available trials are predominantly small-scale studies with limited duration. Moreover, they profoundly differ in tested compounds or composition of extracts or foods, dosage, intervention schemes, population, endpoints, and markers. More randomized, placebo-controlled trials using validated biomarkers, and with sufficient follow-up are needed. Additionally, the polyphenol composition of food is a result of cultivation, processing, storage, and cooking parameters, implicating a large variability. Often, no or inadequate polyphenol composition has been assessed. Furthermore, it is difficult to identify the bioactivity of a single polyphenol, as clinical effects of foodstuffs are likely to be the result of interactions between different polyphenols, and of polyphenols with other food components, interfering with different molecular targets simultaneously.

Additionally, the extensive metabolism of polyphenols adds substantially to the potential bioactive compound array. Most studies do not take into account bioavailability data and levels and activity of polyphenol phase I and II and microbial metabolites, though this is of major importance. Detailed monitoring of polyphenol bioavailability is, therefore, required. Apart from the biological activities of those metabolites, intracellular deconjugation metabolism of phase II metabolites should also be taken into account, releasing parent polyphenols in cells and tissues and provoking local activity. Bidirectional influences between polyphenols and the intestinal microbiome magnify the heterogeneity of data reported in epidemiological or clinical assays. Moreover, gut microbial composition and metabolism itself influence cardiovascular risk, in general, and AS, in particular. Strategies involving targeting the microbiome with probiotics or prebiotics could, therefore, be valuable in arterial stiffness treatment. Also, polyphenols displaying prebiotic-like effects can potentially be used to modulate intestinal microbiota.

Application of metabolomics approaches could identify all polyphenolic metabolites involved in an observed effect and will help to elucidate mechanisms and targets of their activity in arterial stiffness. However, it should be considered that even the use of valid biomarkers and metabolomics can be biased by several factors, including those influencing microbial metabolism.

## Figures and Tables

**Figure 1 nutrients-11-00578-f001:**
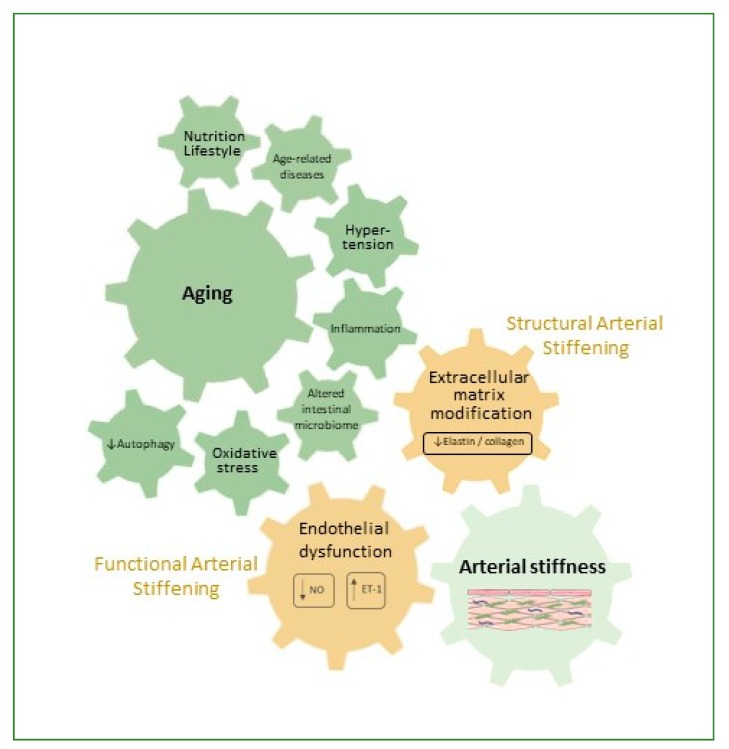
Mechanisms which contribute to the pathophysiology of arterial stiffness. NO: nitric oxide; ET-1: endothelin-1.

**Figure 2 nutrients-11-00578-f002:**
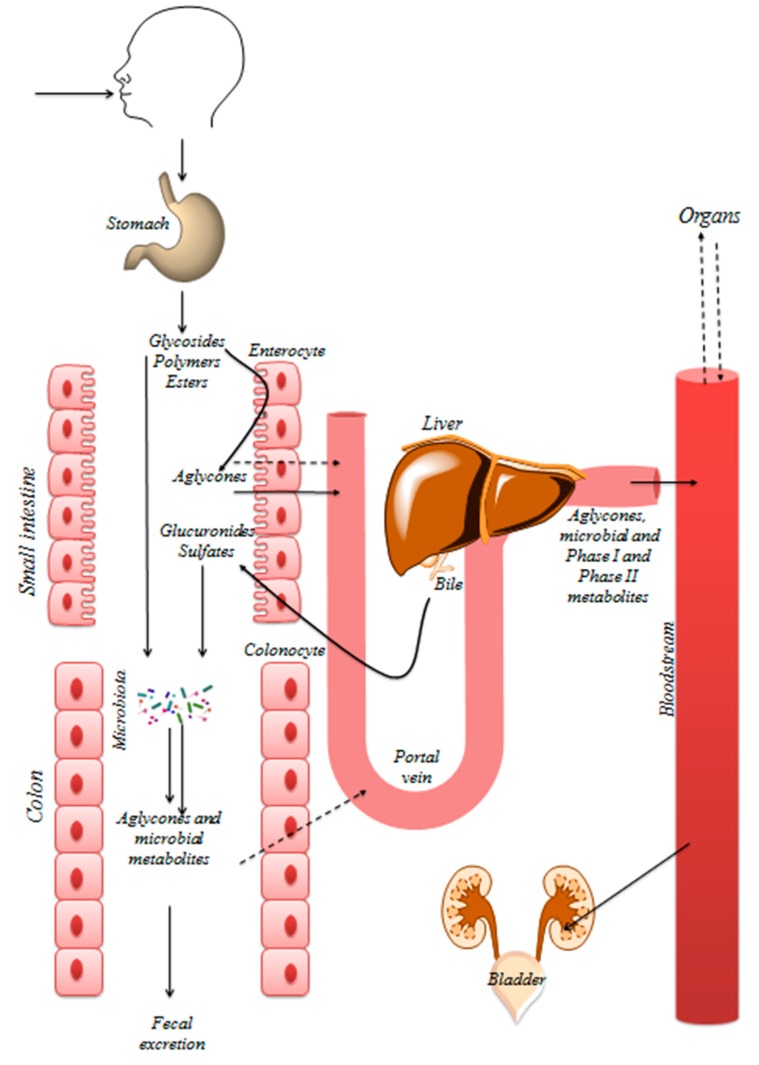
Schematic representation of absorption, biotransformation, and excretion of polyphenols in the human body. The gastrointestinal tract acts as a physical barrier, is covered by the mucosa, and is decisive for polyphenols bioavailability. This function is mediated by physical walls, metabolism, and passive (solid arrows)/active (dashed arrows) transport mechanisms. Polyphenols interact with salivary proteins, but they are not metabolized in the oral cavity. Moreover, most of the polyphenols resist to stomach’s acidic conditions and may be transported bound to dietary plant polysaccharides. Absorption occurs mainly at the duodenum and the proximal half of jejunum at enterocytes. Apical cell membranes of enterocytes contain microvilli, which increase the surface area of absorption. Passive intestinal permeability occurs mainly for aglycones and simple phenolic acids. Absorption of glycosylated compounds is usually preceded by the release of aglycones by enzymes. Free aglycones can then enter the epithelial cells by passive diffusion. Alternatively, glycosylated compounds enter epithelial cells by the active transport and are hydrolyzed by intracellular enzymes. Once inside enterocytes, polyphenols can be extruded into the lumen by efflux transporters. Compounds which are not absorbed reach the colon where they can be extensively metabolized by microbiota. Several transformations in (poly)phenols structure can occur. Most of the colonic metabolites are excreted in feces, although absorption can still take place. Then, (poly)phenols can undergo phase I and phase II reactions. Phase I reactions include oxidative and reductive reactions. Glucuronidation, sulfation, and methylation are the most frequent phase II reactions. The conjugates, being more water-soluble, are rapidly excreted through bile or urine. Metabolites can then be transported into the bile (enterohepatic recirculation) and secreted back to the duodenum. Degradation of metabolites in the intestine generates catabolites available for reabsorption (adapted from [42]).

**Figure 3 nutrients-11-00578-f003:**
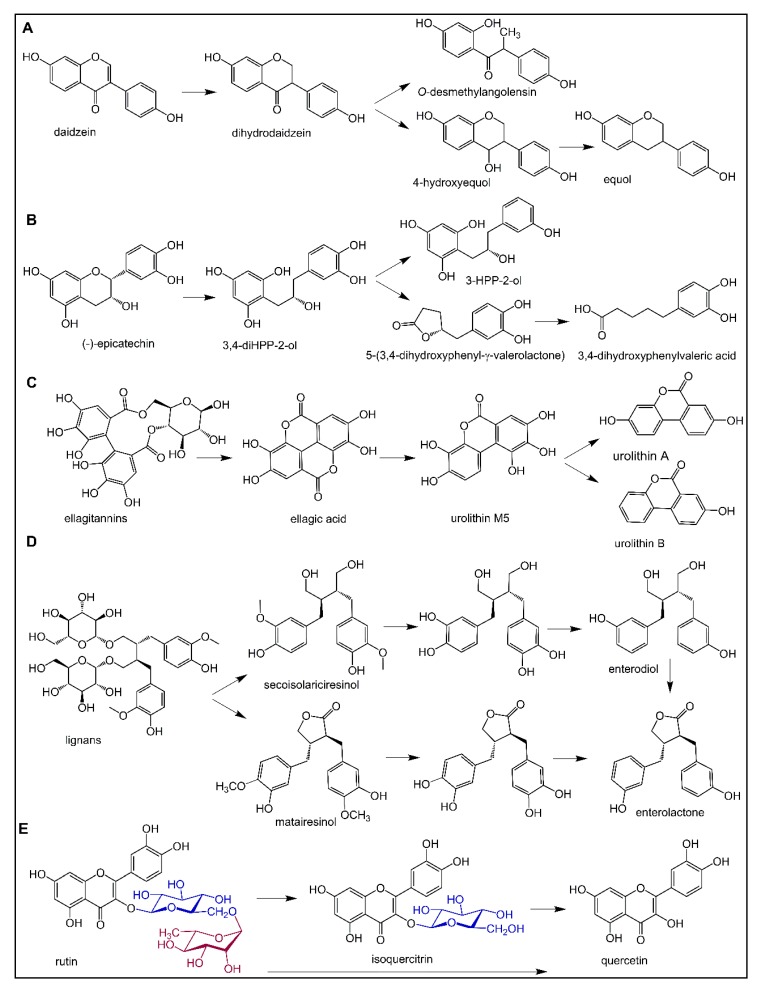
Microbial biotransformation of isoflavonoids (**A**), flavan-3-ols (**B**), ellagitannins (**C**), lignans (**D**), and flavonoid rutinosides (**E**).

**Figure 4 nutrients-11-00578-f004:**
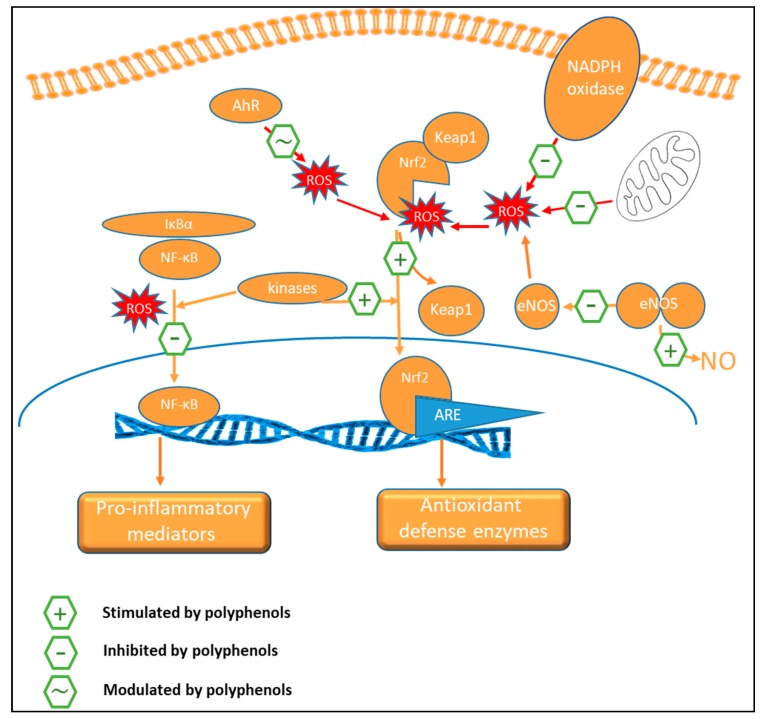
Oxidative/antioxidative pathways involved in arterial stiffness etiology and interactions of dietary polyphenols. Pathways stimulated, inhibited or modulated by polyphenols are indicated by +, − or ~, respectively. AhR: aryl hydrocarbon receptor; ARE: antioxidant response element; eNOS: endothelial nitric oxide synthase; IκBα: nuclear factor of kappa light polypeptide gene enhancer in B-cells inhibitor, alpha; Keap1: Kelch-like ECH-associated protein 1; NADPH oxidase: Nicotinamide adenine dinucleotide phosphate oxidase; NF-κB: nuclear factor kappa-light-chain-enhancer of activated B cells; NO: nitric oxide; Nrf2: Nuclear factor (erythroid-derived 2)-like 2; ROS: reactive oxygen species.

**Figure 5 nutrients-11-00578-f005:**
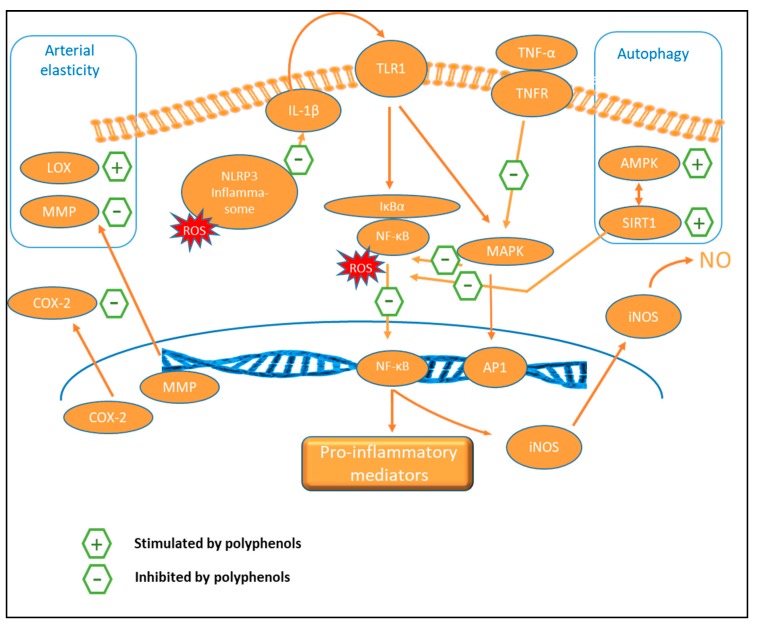
Inflammatory pathways involved in arterial stiffness etiology and interactions of dietary polyphenols. Pathways stimulated or inhibited by polyphenols are indicated by + or −, respectively. AMPK: adenosine monophosphate (AMP)-activated protein kinase; AP1: activator protein 1; COX-2: cyclooxygenase-2; iNOS: inducible nitric oxide synthase; IκBα: nuclear factor of kappa light polypeptide gene enhancer in B-cells inhibitor, alpha; IL-1β: interleukin 1 beta; LOX: lysyl oxidase; MAPK: mitogen-activated protein kinase; MMP: matrix metalloprotease; NF-κB: nuclear factor kappa-light-chain-enhancer of activated B cells; NLRP3: Nod-like receptor protein 3; NO: nitric oxide; ROS: reactive oxygen species; SIRT1: sirtuin 1; TLR1: toll-like receptor 1; TNF-α: Tumor necrosis factor alpha; TNFR: tumor necrosis factor receptor.

**Table 1 nutrients-11-00578-t001:** Dietary polyphenols and their sources.

Dietary Polyphenol Class	Subclass	Compounds (Examples)	Dietary Sources (Examples)
**Phenolic acids**		Chlorogenic acid, caffeic acid, gallic acid, ferulic acid	coffee, berries, kiwi, apple, cherry
**Phenolic alcohols**		Hydroxytyrosol	olive
**Stilbenes**		Resveratrol	grapes, wine
**Lignans**		Secolariciresinol	linseed
**Flavonoids**			
	Isoflavones	Genistein, daidzein	soy, miso
	Flavones	Luteolin, apigenin	celery, parsley, capsicum pepper
	Flavanones	Hesperetin, naringenin	oranges, grapefruit, lemon
	Flavonols	Quercetin, kaempferol, myricetin	onion, leek, broccoli, berries
	Flavanols	(Epi)catechins, (epi)gallocatechins, epigallocatechin gallate	grapes, wine, cocoa, apricots, beans, green tea
	Anthocyanins	Delphinidin, cyanidin, malvidin	berries, aubergine, black grapes, rhubarb, red wine
**Tannins**	Condensed tannins	Procyanidins	cocoa, chocolate, apples, grapes
	Hydrolyzable tannins	Gallotannins, ellagitannins	mango, pomegranate

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
