# Peer review of "Dietary Polyphenols Targeting Arterial Stiffness: Interplay of Contributing Mechanisms and Gut Microbiome-Related Metabolism"

_nutrients, 2019, doi:10.3390/nu11030578_

Round 1
Reviewer 1 Report
This is an interesting and well-written (very long) review on the effect of polyphenols on vascular function. The concepts are well explained and the argument is of clinical interest. Few minor points should be clarified:
1) The reviewer fully agree to the intent of Figure 1. However, it should be improved. Maybe a diagram that include known interrelationship and pathogenesis of arterial stiffening (see for example PMID: 26523102) could be preferable.
2) Can polyphenols and gut microbiome have a role on chronic inflammation and arterial stiffness in patients with chronic inflammation and, in particular in those with inflammatory bowel disease?
3) Can the authors report in a figure potential mechanisms by which polyphenols improve arterial stiffness? Gut microbiome is an emerging cause of arterial stiffening (PMID:29860302). Can the authors explain the role of polyphenols in this process?
4) P10, R395: the authors wrote “Often evaluation of AS markers is combined with registration of effects on BP and endothelial function, parameters that are also affected by vascular stiffness”. Endothelial function can lead to functional arterial stiffening (see PMID: 26523102).
5) Chronic inflammation is an emerging pathogenic factor of arterial stiffening in patients with inflammatory bowel disease, rheumatoid arthritis, systemic lupus erythematosus, etc. Can polyphenols and gut microbiome have a role in this process at least in some of these chronic diseases? Accordingly, anti-tumor necrosis factor therapy leads to the normalization of aortic stiffness in patients with inflammatory bowel disease and in those with rheumatoid arthritis. Can polyphenols have a role in this process? Maybe a figure could be useful.
6) The figures should be self-explaining and abbreviation should be reported in figure legends.
7) Can the authors better report the role of polyphenols in Figure 4-5?
8) Please, check the references: reference 2 and 10 are the same.
Author Response
Dear Reviewer,
We would like to thank you very much for reading our manuscript and for the comments and remarks made. We have uploaded our reply in a Word file.
Yours Sincerely,
Nina Hermans

Reviewer 2 Report
The authors have presented a very extensive review of the effects of dietary polyphenols in targeting arterial stiffness. The review is systematically laid out and has a good flow. The figures are used well to summarize important points. I would suggest the authors include a table listing the dietary polyphenol components and sources.
Author Response
Dear Reviewer,
We would like to thank you very much for reading our manuscript and for the suggestion made. Please find our response here:
The authors have presented a very extensive review of the effects of dietary polyphenols in targeting arterial stiffness. The review is systematically laid out and has a good flow. The figures are used well to summarize important points. I would suggest the authors include a table listing the dietary polyphenol components and sources.
Thank you for your comments. A table covering major dietary polyphenol classes and their most important dietary sources has been included now. (Table 1)
Yours Sincerely,
Nina Hermans